**Data Availability Statement:** All relevant data are within the paper t and its Supporting information files.

# Disparities in behavioral health and experience of violence between cisgender and transgender Thai adolescents

**Wit Wichaidit[1,2]\*, Sawitri Assanangkornchai[1,2], Virasakdi Chongsuvivatwong[1]**

**1** Epidemiology Unit, Faculty of Medicine, Prince of Songkla University, Hat Yai, Thailand, **2** Centre for Alcohol Studies, Hat Yai, Thailand

\* wit.w@psu.ac.th

## Abstract

### Background

The term "transgender" refers to an individual whose gender identity is different from their sex assigned at birth, whereas the term "cisgender" refers to an individual whose gender identity is the same as their sex assigned at birth. In Thailand, studies on health outcomes and quality of life of Thai transgender youths have not included assessments from nationally-representative samples. The objective of this study is to assess the extent that behavioral health outcomes and exposure to violence varied by gender among respondents of the National School Survey on Alcohol Consumption, Substance Use and Other Health-Risk Behaviors.

### Methods

We used data from a nationally-representative self-administered survey of secondary school students in years 7, 9 and 11 and classified participants as cisgender boys, cisgender girls, transgender boys, and transgender girls. Participants also answered questions on depressive experience, suicidality, sexual behaviors, alcohol and tobacco use, drug use, and past-year experience of violence. We analyzed data using descriptive statistics and modified multivariate Poisson regression with adjustment for sampling weights to calculate adjusted prevalence ratios (APR) with 95% confidence intervals.

### Results

A total of 31,898 respondents (82.8% of those who returned complete and valid questionnaires) answered questions on sex and gender identity and were included in the analyses (n = 31,898 respondents), approximately 2.5% of whom identified as transgender. Transgender boys had a higher prevalence suicidal ideation than cisgender boys (APR = 2.97; 95% CI = 1.89, 4.67) and cisgender girls (APR = 2.29; 95% CI = 1.55, 3.40). Transgender girls were less likely than cisgender boys and girls to be ever drinkers, while transgender boys were more likely than cisgender boys and girls to be ever drinkers. Transgender girls had

**Funding:** The National School Survey on Alcohol Consumption, Substance Use and Other Health-Risk Behaviors 2016 was funded by the Thailand Substance Abuse Academic Network (TSAN), Thai Health Promotion Foundation. The secondary data analysis was funded by the Centre for Alcohol Studies, Thai Health Promotion Foundation. The funders had no role in study design, data collection and analysis, decision to publish, or preparation of the manuscript.

**Competing interests:** The authors have declared that no competing interests exist.

higher past-year exposure to sexual violence than cisgender boys (APR = 2.74; 95% CI = 1.52, 4.95) and cisgender girls (APR = 4.93; 95% CI = 2.52, 9.67).

## Conclusion

We found disparities in behavioral health and experience of violence between transgender and cisgender adolescents in Thailand. The findings highlighted the need for program managers and policy makers to consider expanding local efforts to address health gaps in the LGBTQ community to also include school-going youth population.

## Introduction

Gender spectrum refers to the idea that gender can be seen as a fluid, multidimensional concept, where gender exists anywhere on a continuum that includes male and female [1]. Gender identity development refers to the development of the extent to which a person identifies oneself within the gender spectrum [2]. Gender identity gradually develops, starting from early childhood [2] and is complete around the time of puberty [3]. The term "cisgender" refers to an individual whose gender identity is the same as their sex assigned at birth. The term "transgender" refers to an individual whose gender identity is different from their sex assigned at birth.

Adolescence is also a precarious period when one is susceptible to behavioral health issues that can affect health outcomes in later life [4]. Behavioral health issues affect both transgender and cisgender adolescents, although a previous study has shown disparities in the prevalence of unsafe sexual practices, sexual violence, drug use, and depression [5]. Transgender people tend to have higher prevalence of high-risk behavior [6, 7], depression [8] and suicidality [8, 9].

Thailand is a middle-income country in South East Asia that experiences homonegativity, not unlike other countries in the region [10]. Although many perceive the country as a safe haven [11], Thai transgender women do report experiences of discrimination and violence [12]. Studies on health outcomes and quality of life of Thai transgender youths are either include only qualitative methods [13], or are mostly based on small-scale surveys without assessment of disparities between transgender and cisgender youths [14]. Moreover, previous studies tend to focus on male-to-female transgender youths, and scarce data exists on the health of female-to-male transgender youths.

The National School Survey on Alcohol Consumption, Substance Use and Other Health-Risk Behaviors was a nationally-representative large-scale cross-sectional study among Thai school-going adolescents that included questions on gender identity and behavioral health, presenting an opportunity to assess disparities by gender [15]. Secondary data from the survey are available for analysis. The 2016 survey was the third of its kind (the first two surveys were conducted in 2007 and 2009) and was the first survey in which the gender identity question was included in addition to the sex assigned at birth question. It is thus possible to assess disparities in behavioral health outcomes and experience of violence between Thai cisgender and transgender adolescents. Such information can provide evidence to inform school health programs regarding disparities in health problems and need and to plan and allocate resources for programmatic efforts to better support student health. The objective of this study is to assess the extent that behavioral health outcomes and exposure to violence varied by gender among

respondents of the National School Survey on Alcohol Consumption, Substance Use and Other Health-Risk Behaviors.

## Materials and methods

### Study design and participants

The National School Survey on Alcohol Consumption, Substance Use and Other Health-Risk Behaviors (hence "Survey") was a cross-sectional survey conducted to provide information about the magnitude and trend of alcohol consumption, tobacco use, substance use and other high-risk behaviors among Thai adolescents in Thailand's formal educational system. The participants included students in Year 7 (*Matthayom 1*), Year 9 (*Matthayom 3*), and Year 11 in the general education system (*Matthayom 5*) and the vocational education system (Vocational Certificate Year 2). The survey covered public and private schools in urban and rural areas, and 40 out of 77 provinces of Thailand.

### Instrumentation: Measurement of gender

We used the responses of two questions to define the genders in this study (Sub-Section A1: Sex): sex assigned at birth, and gender identity. The first question in Sub-Section A1 measured sex assigned at birth with the question "*A1.1 Sex at Birth*" with two possible answer choices: "1) Male", and "2) Female". The second question in Sub-Section A1 measured gender identity with the question "*A1.2 Gender Identity (you think that your gender is. . .)*" with two possible answer choices: "1) Male", and "2) Female".

Among the respondents, 0.9% did not answer A1.1 (sex assigned at birth) and 17.7% did not answer A1.2 (gender identity). We excluded participants who did not answer either one of these questions from the analysis. We therefore categorized students into 4 groups: 1) students who answered "Male" in A1.1 and "Male" in A1.2, labelled as "Cisgender Boys"; 2) students who answered "Female" in A1.1 and "Female" in A1.2, labelled as "Cisgender Girls"; 3) students who answered "Male" in A1.1 and "Female" in A1.2, labelled as "Transgender Girls"; 4) students who answered "Female" in A1.1 and "Male" in A1.2, labelled as "Transgender Boys" (Table 1).

### Instrumentation: Behavioral health

Behavioral health outcomes in this survey included depressive experience, suicidality, unsafe sexual behaviors, alcohol consumption, tobacco consumption, and drug use. Depressive experience referred to history of feeling of sadness or despair on a near-daily basis for a period of two weeks or more within the past 12 months. Suicidality-related questions included history of

**Table 1. Gender classification definitions of respondents and proportions, weighted percent ± standard error (SE) (n = 31,898 respondents).**

| Group | Definition | Percent ± SE |
|---|---|---|
| **1) Cisgender Boys (n = 14,040 respondents)** | Respondents whose sex was "Male" and gender identity was "Male" | 42.5% ± 1.3% |
| **2) Cisgender Girls (n = 17,103 respondents)** | Respondents whose sex was "Female" and gender identity was "Female" | 55.0% ± 1.2% |
| **3) Transgender Girls (n = 421 respondents)** | Respondents whose sex was "Male" and gender identity was "Female" | 1.4% ± 0.1% |
| **4) Transgender Boys (n = 334 respondents)** | Respondents whose sex was "Female" and gender identity was "Male" | 1.1% ± 0.1% |

suicidal ideation, suicide planning, and suicide attempts within the past 12 months. Non-responses were treated as missing data.

Unsafe sexual behaviors included lifetime history of sexual intercourse, use of alcohol during last sexual encounter, use of illicit drugs during last sexual encounter, and use of contraception by respondent or partner during the last sexual encounter. We defined a lack of contraceptive use as reporting "None" or "Withdrawal (*coitus interruptus*)" as the method for contraception. Analyses of data related to sexual behaviors other than lifetime history were limited to only students who reported a lifetime history of sexual intercourse. We excluded students who did not answer the "have you ever had sex" question from the analyses.

For drinking and smoking behaviors, we considered students who reported ever drinking in their lifetime to be ever drinkers, and students who reported drinking in the past 12 months to be current drinkers. Similarly, we considered students who reported having smoked 100 cigarettes or more in their lifetime to be ever smokers, and students who reported smoking in the past 12 months to be current smokers. We excluded students who did not provide an answer on ever-drinking from the analyses on drinking, and we excluded students who did not provide an answer on ever-smoking from the analyses on smoking. We also assumed that students who never drank also had not consumed alcohol within the past 12 months, and that students who never smoked also had not smoked within the prior 12 months. The 12-month time frame for alcohol consumption was chosen to be consistent with the Tobacco and Alcohol Consumption Survey 2017 [16]. The 12-month time frame for tobacco use was chosen to be consistent with the definition used in the Global Adult Tobacco Survey (GATS) [17].

For drug use behaviors, we included drugs where the lifetime history of use was higher than one percent in the entire population: 1) marijuana; 2) *kratom* (*Mitragyna speciosa*); 3) methamphetamine pills, and; 4) crystal methamphetamine. The one percent cut-point was arbitrarily chosen in order to achieve adequate statistical power to make comparisons between cisgender and transgender youths. Non-responses were treated as missing data.

## Instrumentation: Measurement of experience of violence

Exposure to violence was measured with questions on experience within the past 12 months of the survey regarding: 1) being threatened or assaulted with weapons; 2) being in a fight or quarrel; 3) intimate partner violence; 4) sexual violence. For intimate partner violence and sexual violence, we excluded students who never had an intimate partner and students who never had sex from the analyses, respectively. We also excluded non-responses from the analyses. We excluded participants who did not report any of the four types of violence from the calculation of exposure to any type of violence in the past 12 months.

## Instrumentation: Demographic characteristics

The General Characteristics Section of the questionnaire contained information on school type, year level and track of study, religion, geographic region, living situation (participant's accommodation arrangement), monthly allowance, and grade point average (GPA). We described variations regarding these characteristics by gender identities and included these as co-variables during multivariate analyses.

## Procedure

The survey included data from schools in 40 out of 77 provinces. The survey researchers randomly selected half of the provinces in each of the 12 education regions (plus two districts in the capital of Bangkok), then sampled the schools. Survey researchers then contacted the school administrtors to ask for permission to conduct the Survey, and enumerators then

visited the school, distributed the self-administered questionnaires to the students, explained about the Survey, and asked the students for verbal consent. The students then completed the questionnaires and placed the questionnaires in individual envelopes. The sampling method and data collection procedures have been described in detail elsewhere [15, 18].

## Data analysis

We performed univariate analyses by describing the basic characteristics of the study respondents. We then performed bivariate analyses by cross-tabulation of gender classification of study respondents and the outcomes (behavioral health and exposure to violence). We then performed multivariate analyses to measure the association between gender classification and the outcomes adjusting for the effect of basic characteristics of the respondents using modified Poisson regression. The analyses were performed separately for each of the outcome variables. Our measure of association was the adjusted prevalence ratio with 95% confidence intervals (APR with 95% CI). In a similar manner to previous studies on health disparities among transgender people [19–21], we compared the prevalence of each outcome in each transgender group to the prevalence among cisgender boys and cisgender girls separately. We adjusted for the respondents' personal characteristics including school type, year level, religion, geographic region, living situation, monthly allowance, and grade point average in the modified Poisson regression analyses, as was similarly done in a previous study [22]. We conducted all data analyses using R with the epicalc package [23] and the survey package [24] to generate weighted estimates and account for the complex study design.

## Ethical considerations

The 2016 national school survey was approved by Khon Kaen University Ethical Review Board for Research in Human Subjects (EC: 59-396-18-1, Project Number HE581430). Ethical clearance for this secondary data analysis was granted exemption by the Ethics Committee for Research in Human Subjects of the Faculty of Medicine, Prince of Songkla University (Project number REC.62-054-18-1). A waiver of the need for a document of consent for minors was approved from the institutional review board. As anonymity was guaranteed and research procedures entailed no more than minimal risk to subjects, students were allowed to provide consent verbally in lieu of signing written informed consent forms.

## Results

One percent (1%) of all students in the sampled classrooms refused to participate in the survey. Among students who agreed to participate (i.e., the respondents), fewer than five percent (5%) returned incomplete or potentially invalid questionnaires which were discarded. Out of 38,535 respondents who returned complete and valid questionnaires, 31,898 (82.8%) answered both questions on sex and gender identity and were included in the analyses (i.e., total sample size = 38,535 respondents; final analytic sample size = 31,898 respondents). Transgender girls and transgender boys accounted for approximately 2.5% of all respondents (Table 1). Respondents of all genders were similar to one another with regards to school type, religion, and geographic region (Table 2). However, transgender girls had higher mean monthly allowance than respondents of other genders, and a higher proportion of those in the highest GPA range (GPA = 3.1 to 4.0) than other genders except cisgender girls. Transgender boys had the highest proportion of respondents who lived in rented houses (either alone or with housemates) and the lowest proportion of respondents who lived in their family's house/flat.

Transgender boys had higher prevalence of depressive experience and suicidality compared to cisgender boys and girls, particularly with regard to suicidal ideation in past 12 months

**Table 2.  Characteristics of study respondents, weighted percent ± SE unless otherwise noted (n = 31,898 respondents).**

| Characteristic | Cisgender boys (Percent ± SE) | Cisgender girls (Percent ± SE) | Transgender girls (Percent ± SE) | Transgender boys (Percent ± SE) | P-value[a] |
|---|---|---|---|---|---|
| | (n = 14,040) | (n = 17,103) | (n = 421) | (n = 334) | |
| **School type** | | | | | |
| Government | 66.7 ± 2.8 | 66.9 ± 3.1 | 73.7 ± 3.4 | 69.2 ± 8.1 | 0.725 |
| Private | 32.4 ± 2.8 | 32.0 ± 3.2 | 25.7 ± 3.4 | 29.8 ± 8.2 | |
| Unknown | 1.0 ± 0.2 | 1.1 ± 0.4 | 0.5 ± 0.4 | 1.0 ± 0.8 | |
| **Year Level** | | | | | |
| Mathayom 1 (Year 7) | 29.0 ± 1.2 | 25.5 ± 1.0 | 12.6 ± 2.1 | 20.1 ± 2.3 | <0.001 |
| Mathayom 3 (Year 9) | 27.3 ± 1.1 | 26.7 ± 0.8 | 29.4 ± 3.3 | 29.0 ± 3.9 | |
| Mathayom 5 (Year 11, General Education) | 23.8 ± 1.1 | 36.0 ± 1.0 | 43.7 ± 3.8 | 29.4 ± 3.2 | |
| Vocational Certificate 2 (Year 11, Vocational Education) | 20.0 ± 1.6 | 11.8 ± 0.9 | 14.3 ± 2.6 | 21.5 ± 2.7 | |
| **Religion** | | | | | |
| Buddhism | 90.6 ± 2.8 | 91.2 ± 2.5 | 86.9 ± 2.8 | 92.0 ± 2.2 | 0.602 |
| Islam | 7.1 ± 2.9 | 6.5 ± 2.7 | 8.3 ± 2.6 | 5.3 ± 2.1 | |
| Christianity | 2.1 ± 0.4 | 2.2 ± 0.5 | 4.5 ± 1.6 | 2.6 ± 0.9 | |
| Others | 0.2 ± 0.0 | 0.1 ± 0.1 | 0.3 ± 0.3 | 0.1 ± 0.1 | |
| **Region** | | | | | |
| Special-Bangkok | 7.8 ± 7.4 | 10.8 ± 9.9 | 13.7 ± 12.2 | 16.5 ± 14.2 | 0.071 |
| Bangkok Metro Areas | 4.4 ± 3.2 | 3.7 ± 2.7 | 4.2 ± 3.0 | 3.9 ± 3.0 | |
| Central | 22.0 ± 7.4 | 21.4 ± 7.9 | 15.6 ± 6.0 | 19.8 ± 7.6 | |
| South | 14.2 ± 6.3 | 13.3 ± 5.7 | 7.8 ± 3.5 | 12.4 ± 6.0 | |
| North | 22.7 ± 7.6 | 23.4 ± 8.2 | 22.8 ± 9.1 | 22.0 ± 8.8 | |
| Northeast | 28.9 ± 8.6 | 27.4 ± 8.7 | 36.0 ± 11.1 | 25.5 ± 9.0 | |
| **Living Situation** | | | | | |
| Family house/flat | 85.1 ± 1.9 | 85.3 ± 1.6 | 80.6 ± 3.2 | 77.7 ± 2.7 | **0.019** |
| School dorm | 4.2 ± 2.0 | 3.3 ± 1.3 | 4.1 ± 2.0 | 2.5 ± 1.1 | |
| Outside dorm | 3.2 ± 0.5 | 2.8 ± 0.5 | 5.5 ± 1.4 | 6.5 ± 2.1 | |
| Rented house | 6.6 ± 1.0 | 7.4 ± 1.4 | 8.2 ± 3.3 | 11.1 ± 2.3 | |
| Others (relatives, temple) | 1.0 ± 0.2 | 1.2 ± 0.3 | 1.6 ± 0.6 | 2.3 ± 1.1 | |
| **Monthly allowance (THB) (mean ± standard errors)** | 2,544.1 ± 93.5 | 2,461.9 ± 110.4 | 3,533.4 ± 450.4 | 3,141.4 ± 440.4 | <0.001[b] |
| **Grade point average (GPA)** | | | | | |
| GPA = 0.1–1.0 | 0.7 ± 0.2 | 0.2 ± 0.1 | 0.8 ± 0.4 | 0.0 ± 0.0 | <0.001 |
| GPA = 1.1–2.0 | 11.3 ± 0.9 | 3.6 ± 0.4 | 6.8 ± 1.5 | 5.2 ± 1.6 | |
| GPA = 2.1–3.0 | 44.2 ± 1.1 | 32.9 ± 1.4 | 38.6 ± 3.1 | 42.0 ± 3.0 | |
| GPA = 3.1–4.0 | 34.7 ± 1.9 | 56.8 ± 1.9 | 49.1 ± 2.7 | 42.6 ± 3.7 | |
| Unknown | 9.0 ± 0.8 | 6.5 ± 0.8 | 4.7 ± 1.4 | 10.2 ± 2.3 | |

[a]Chi-square test of association with Rao & Scott adjustment, unless otherwise noted.

[b]From results of one-way ANOVA with adjustment for complex survey design.

(APR = 2.97; 95% CI = 1.89, 4.67 when compared to cisgender boys, and APR = 2.29; 95% CI = 1.55, 3.40 when compared to cisgender girls) (Table 3). Similarly, transgender girls had higher prevalence of suicidality compared to cisgender boys. Transgender respondents (both boys and girls) also had higher prevalence of foregoing contraceptive use at last sexual encounter compared to cisgender respondents (both boys and girls). Transgender girls were less likely

**Table 3. Prevalence (weighted percent ± SE) and adjusted prevalence ratios (APRs) of mental health outcomes and health behaviors among respondents, stratified by gender.**

| | Cisgender boys (Percent ± SE) | APR (95% CI)[a] | Cisgender girls (Percent ± SE) | APR (95% CI)[a] | Transgender girls (Percent ± SE) | APR (95% CI)[a] (*Ref.* Cisgender boys) | APR (95% CI)[a] (*Ref.* Cisgender girls) | Transgender boys (Percent ± SE) | APR (95% CI)[a] (*Ref.* Cisgender boys) | APR (95% CI)[a] (*Ref.* Cisgender girls) |
|---|---|---|---|---|---|---|---|---|---|---|
| | (n = 14,040) | | (n = 17,103) | | (n = 421) | | | (n = 334) | | |
| *Mental Health in past 12 months* | | | | | | | | | | |
| Depressive experience | (n = 13,004) 12.2% ± 0.6% | 1.0 (Ref.) | (n = 16,346) 14.0% ± 0.8% | 1.0 (Ref.) | (n = 399)19.1% ± 2.3% | 1.31 (0.97, 1.76) | 1.12 (0.82, 1.53) | (n = 313)22.6% ± 3.6% | **1.78 (1.33, 2.40)** | **1.53 (1.15, 2.02)** |
| Suicidal ideation | (n = 9,354) 5.8% ± 0.6% | 1.0 (Ref.) | (n = 11,538) 7.2% ± 0.7% | 1.0 (Ref.) | (n = 290)11.5% ± 2.0% | **1.66 (1.13, 2.45)** | 1.28 (0.89, 1.86) | (n = 239)16.0% ± 3.2% | **2.97 (1.89, 4.67)** | **2.29 (1.55, 3.40)** |
| Suicide planning | (n = 13,152) 4.6% ± 0.3% | 1.0 (Ref.) | (n = 16,487) 5.7% ± 0.4% | 1.0 (Ref.) | (n = 403)9.1% ± 1.7% | **1.69 (1.09, 2.61)** | 1.27 (0.83, 1.94) | (n = 318)13.9% ± 2.4% | **2.82 (1.74, 4.58)** | **2.13 (1.43, 3.17)** |
| Suicide attempt | (n = 13,501) 4.5% ± 0.6% | 1.0 (Ref.) | (n = 16,742) 5.1% ± 0.5% | 1.0 (Ref.) | (n = 416)8.2% ± 2.0% | **1.63 (1.07, 2.49)** | 1.33 (0.90, 1.98) | (n = 327)8.8% ± 1.9% | **2.12 (1.17, 3.85)** | **1.73 (1.01, 2.97)** |
| *Sexual Behaviors* | | | | | | | | | | |
| Ever had sex | (n = 13,618) 17.3% ± 0.8% | 1.0 (Ref.) | (n = 16,676) 10.8% ± 0.8% | 1.0 (Ref.) | (n = 417)21.8% ± 4.1% | 1.09 (0.80, 1.47) | **1.42 (1.04, 1.95)** | (n = 326)15.7% ± 2.9% | 0.88 (0.64, 1.21) | 1.15 (0.83, 1.58) |
| *Among those who ever had sex* | | | | | | | | | | |
| Use of alcohol during last sexual encounter | (n = 2,265) 18.0% ± 1.0% | 1.0 (Ref.) | (n = 1,776) 14.7% ± 1.4% | 1.0 (Ref.) | (n = 76) 24.4% ± 5.4% | 1.05 (0.60, 1.84) | 1.36 (0.69, 2.68) | (n = 48) 14.7% ± 6.4% | 0.75 (0.30, 1.85) | 0.96 (0.44, 2.13) |
| Use of illicit drug during last sexual encounter | (n = 2,247) 6.9% ± 0.7% | 1.0 (Ref.) | (n = 1,760) 3.3% ± 0.5% | 1.0 (Ref.) | (n = 71) 5.9% ± 3.0% | 0.41 (0.33, 0.60) | 0.92 (0.26, 3.29) | (n = 48) 3.6% ± 2.6% | 0.57 (0.12, 1.45) | 1.28 (0.27, 6.00) |
| Foregoing contraceptive use during last sexual encounter | (n = 2,258) 28.2% ± 1.7% | 1.0 (Ref.) | (n = 1,766) 25.6% ± 1.3% | 1.0 (Ref.) | (n = 72) 47.2% ± 7.5% | **1.96 (1.39, 2.76)** | **2.11 (1.59, 2.80)** | (n = 49) 37.5% ± 8.5% | **1.60 (1.05, 2.43)** | **1.72 (1.14, 2.61)** |
| Condom use during last sexual encounter | (n = 2,258) 57.4% ± 1.8% | 1.0 (Ref.) | (n = 1,766) 51.1% ± 1.5% | 1.0 (Ref.) | (n = 72) 40.0% ± 8.3% | 0.59 (0.33, 1.06) | 0.68 (0.39, 1.17) | (n = 49) 19.1% ± 8.7% | **0.34 (0.14, 0.81)** | **0.39 (0.16, 0.92)** |
| *Alcohol and tobacco use* | | | | | | | | | | |
| Ever drinker | (n = 13,706) 39.3% ± 1.1% | 1.0 (Ref.) | (n = 16,786) 35.6% ± 1.6% | 1.0 (Ref.) | (n = 412)33.6% ± 2.7% | **0.81 (0.70, 0.95)** | 0.85 (0.73, 1.00) | (n = 331)51.1% ± 5.2% | **1.27 (1.06, 1.51)** | **1.33 (1.12, 1.59)** |
| Drank in past 12 months (among ever drinkers) | (n = 5,162) 77.0% ± 1.2% | 1.0 (Ref.) | (n = 5,686) 73.6% ± 1.0% | 1.0 (Ref.) | (n = 136)79.1% ± 4.6% | 1.01 (0.91, 1.11) | 1.04 (0.94, 1.16) | (n = 163)75.1% ± 3.2% | 0.97 (0.90, 1.05) | 1.01 (0.93, 1.09) |
| Ever smoker | (n = 13,766) 23.4% ± 1.8% | 1.0 (Ref.) | (n = 16,743) 6.1% ± 0.9% | 1.0 (Ref.) | (n = 410)13.2% ± 3.1% | **0.48 (0.34, 0.67)** | **1.46 (1.04, 2.05)** | (n = 329)20.9% ± 4.2% | 0.91 (0.66, 1.24) | **2.77 (1.90, 4.03)** |
| Smoked in past 12 months (among ever smokers) | (n = 3,523) 74.4% ± 1.6% | 1.0 (Ref.) | (n = 1,120) 70.3% ± 2.4% | 1.0 (Ref.) | (n = 52) 67.0% ± 4.6% | **0.82 (0.60, 1.12)** | 0.85 (0.61, 1.18) | (n = 80) 56.9% ± 5.4% | **0.74 (0.60, 0.91)** | 0.76 (0.61, 0.94) |
| *Lifetime history of illicit drug use* | | | | | | | | | | |
| Marijuana | (n = 10,746) 8.1% ± 1.0% | 1.0 (Ref.) | (n = 13,383) 2.9% ± 0.4% | 1.0 (Ref.) | (n = 284)6.2% ± 2.5% | 0.56 (0.24, 1.30) | 1.43 (0.63, 3.21) | (n = 261)8.9% ± 1.7% | 1.01 (0.66, 1.55) | **2.58 (1.80, 3.69)** |
| Kratom | (n = 10,667) 6.0% ± 1.0% | 1.0 (Ref.) | (n = 13,382) 2.2% ± 0.3% | 1.0 (Ref.) | (n = 283)4.9% ± 2.2% | 0.64 (0.30, 1.39) | 1.44 (0.65, 3.17) | (n = 260)6.5% ± 1.3% | 1.17 (0.83, 1.65) | **2.61 (1.64, 4.16)** |
| Yaba (methamphetamine pills) | (n = 10,673) 2.9% ± 0.6% | 1.0 (Ref.) | (n = 13,369) 1.0% ± 0.3% | 1.0 (Ref.) | (n = 279)0.7% ± 0.5% | 0.10 (0.01, 1.07) | 0.29 (0.03, 2.55) | (n = 261)3.4% ± 1.3% | 1.07 (0.42, 2.73) | **2.99 (1.33, 6.73)** |

*(Continued)*

**Table 3.** (Continued)

| | Cisgender boys (Percent ± SE) | APR (95% CI)[a] | Cisgender girls (Percent ± SE) | APR (95% CI)[a] | Transgender girls (Percent ± SE) | APR (95% CI)[a] (*Ref.* Cisgender boys) | APR (95% CI)[a] (*Ref.* Cisgender girls) | Transgender boys (Percent ± SE) | APR (95% CI)[a] (*Ref.* Cisgender boys) | APR (95% CI)[a] (*Ref.* Cisgender girls) |
|---|---|---|---|---|---|---|---|---|---|---|
| | (n = 14,040) | | (n = 17,103) | | (n = 421) | | | (n = 334) | | |
| Ice (crystal methamphetamine) | (n = 10,639) 1.9% ± 0.4% | 1.0 (Ref.) | (n = 13,360) 1.0% ± 0.2% | 1.0 (Ref.) | (n = 278)1.9% ± 0.8% | 0.60 (0.15, 2.31) | 1.17 (0.31, 4.40) | (n = 260)1.8% ± 0.8% | 0.81 (0.29, 2.23) | 1.58 (0.58, 4.30) |

[a]Adjusted for school type, year level, religion, geographic region, living situation, monthly allowance, and grade point average. Bold texts denote statistically significant association.

than cisgender boys and girls to be ever drinkers, while transgender boys were more likely than cisgender boys and girls to be ever drinkers. Similar differences were also observed in prevalence of being ever smokers. Transgender boys had significantly higher prevalence of life-time history of using illicit drugs compared to cisgender girls, particularly in the use of yaba (methamphetamine pills) (APR = 2.99; 95% CI = 1.33, 6.73).

Both transgender girls and transgender boys had higher exposure to violence in the past year than cisgender girls (Table 4), including being threatened, severe physical violence, and intimate partner violence. Transgender girls had higher past-year exposure to sexual violence than all other groups (APR = 2.74; 95% CI = 1.52, 4.95 compared to cisgender boys, and APR = 4.93; 95% CI = 2.52, 9.67 compared to cisgender girls).

## Discussion

In a nationally-representative survey, we compared behavioral risk factors and past-year experience of violence between Thai transgender and cisgender secondary school students. We

**Table 4. Prevalence (weighted percent ± SE) and adjusted odds ratios (AORs) of past-year exposure to violence among respondents of the TSSHRBS, stratified by gender.**

| | Cisgender boys (Percent ± SE) | APR (95% CI)[a] | Cisgender girls (Percent ± SE) | APR (95% CI)[a] | Transgender girls (Percent ± SE) | APR (95% CI)[a] (*Ref.* Cisgender boys) | APR (95% CI)[a] (*Ref.* Cisgender girls) | Transgender boys (Percent ± SE) | APR (95% CI)[a] (*Ref.* Cisgender boys) | APR (95% CI)[a] (*Ref.* Cisgender girls) |
|---|---|---|---|---|---|---|---|---|---|---|
| | (n = 14,040) | | (n = 17,103) | | (n = 421) | | | (n = 334) | | |
| *Past-year exposure to violence* | | | | | | | | | | |
| Being threatened | (n = 13,366) 8.0% ± 0.6% | 1.0 (Ref.) | (n = 16,611) 2.0% ± 0.1% | 1.0 (Ref.) | (n = 400) 6.7% ± 1.3% | 0.66 (0.40, 1.09) | **2.37 (1.48, 3.82)** | (n = 317) 6.4% ± 2.1% | 0.71 (0.43, 1.14) | **2.52 (1.37, 4.65)** |
| Severe physical violence | (n = 13,373) 7.6% ± 0.5% | 1.0 (Ref.) | (n = 16,613) 2.5% ± 0.1% | 1.0 (Ref.) | (n = 401) 7.9% ± 1.5% | 0.91 (0.54, 1.52) | **2.33 (1.48, 3.68)** | (n = 318) 7.3% ± 2.0% | 0.89 (0.62, 1.28) | **2.29 (1.63, 3.23)** |
| Intimate partner violence | (n = 13,192) 7.3% ± 0.4% | 1.0 (Ref.) | (n = 16,452) 3.0% ± 0.3% | 1.0 (Ref.) | (n = 397) 5.0% ± 1.2% | 0.58 (0.34, 0.97) | 1.16 (0.68, 1.98) | (n = 313) 10.8% ± 2.8% | 1.33 (0.99, 1.78) | **2.66 (2.09, 3.38)** |
| Sexual violence | (n = 13,241) 2.9% ± 0.2% | 1.0 (Ref.) | (n = 16,531) 1.3% ± 0.1% | 1.0 (Ref.) | (n = 405) 9.4% ± 4.3% | **2.74 (1.52, 4.95)** | **4.93 (2.52, 9.67)** | (n = 319) 4.1% ± 1.2% | 1.28 (0.74, 2.23) | **2.31 (1.30, 4.12)** |
| Experienced any type of violence in past 12 months | (n = 13,664) 15.5% ± 0.9% | 1.0 (Ref.) | (n = 16,843) 6.4% ± 0.3% | 1.0 (Ref.) | (n = 408) 18.6% ± 4.4% | 1.04 (0.78, 1.38) | **2.22 (1.68, 2.94)** | (n = 330) 17.0% ± 2.9% | 0.98 (0.71, 1.35) | **2.09 (1.54, 2.83)** |

[a]Adjusted for school type, year level, religion, geographic region, living situation, monthly allowance, and grade point average. Bold texts denote statistically significant association.

found significant disparities between transgender and cisgender youths. We found that transgender boys had the highest levels of depressive experience, suicidality, and history of alcohol consumption compared to other groups, while transgender girls had the highest prevalence of experiencing sexual violence within the year prior to the survey.

Parent or guardian permission was not a requirement for participation in this study, which might have improved the willingness of transgender youths to participate and identify themselves in the survey [25]. However, approximately one-sixth of the respondents did not answer the gender identity question. The Thai word in the questionnaire was '*phet withi*', which was not part of vernacular Thai. It is possible that respondents who did not answer the gender identity question either did not understand it or perceived it to be a duplicate of the question pertaining to sex assigned at birth and decided to skip it. Future studies should consider changing the sex assigned at birth question may be changed from "Sex. . ." to "What sex were you assigned at birth?". Future studies should consider modification of the gender identity measurement question to help reduce this non-response, e.g., changing from "You think you are. . ." ("*Khun kid waa khun ben phet. . .*") to "What gender do you identify as?" ("*Tuaton tii tae jing khun ben phet dai*") to reflect the notion that gender identity is firmly felt and integral to one's being. With regard to the answer choices, both the birth gender and gender identity questions contained only binary responses of 'male' and 'female'. The responses to the gender identity question may include additional answers of 'Not sure / Questioning', 'Genderfluid', 'Non-binary', and 'Do not identify as male, female, or transgender' in order to allow respondents to identify themselves as questioning, genderfluid and gender non-binary. Youths who are in the developmental stage may be questioning their gender identity and affectional orientation [26]. Adding the option of 'Not sure' to the gender identity question may have allowed us to capture "Questioning youths" as a distinct group. The actual responses, however, may be further tailored to suit the context and culture of the study setting and include even more answer choices [27]. In addition, we only asked the respondents about their sex and gender identity, and not their sexual orientation. In that regard, cisgender respondents who were homosexuals and bisexuals were identified in the same group as cisgender respondents who were heterosexual. Likewise, transgender participants were also presumably grouped together without regard for their sexual identity. Future surveys should include a separate question to identify sexual orientation.

Transgender girls had a higher GPA distribution than other groups except for cisgender girls. There has been no empirical study on the underlying reason behind this distinction, but anecdotes suggest that Thai transgender women try to gain acceptance from society by educational and professional achievements [28, 29]. The relatively high GPA distribution among transgender girls could be a reflection of these endeavors during adolescence. In that regard, the literatures also suggest that Thai transgender people face discrimination and stigma in the educational system [12, 30]. Policy makers should continue to encourage academic achievements among Thai transgender youths while ensuring that youths are protected from discrimination and stigma in the educational system.

The study findings showed concerning disparities in behavioral health and experience of violence between transgender and cisgender youths. Suicidality was higher among transgender respondents than cisgender respondents, and was higher among transgender boys than transgender girls. Depression and suicidality among Thai transgender youths are associated with victimization and internalized homophobia [8, 9], as well as family rejection and social isolation [31]. The Thai media also contains hetero-sexist narratives, harmful and discriminatory rhetoric, and negative portrayals of LGBTIQ people [32]. A previous study found minority stress (i.e., stress faced by members of stigmatized minority groups caused by factors such as lack of social support, low socioeconomic status, interpersonal prejudice, and discrimination)

in the Thai homosexual and bisexual men population [33]. It is possible that minority stress and internalization of rejection, discrimination and violence [9, 34] could have accounted for disparities in depression and suicidality between Thai cisgender and transgender adolescents.

With regard to drinking, transgender boys had a higher prevalence of alcohol consumption than cisgender boys, while the prevalence was lower among transgender girls. The reasons for these differences are unclear. However, it is possible that as the majority of drinkers are men [16] and drinking is perceived as a masculine activity [35], transgender boys might have felt compelled to engage in drinking or to conform to their identified gender. In addition, anecdotes suggested that Thai transgender women believe that the simultaneous use of alcohol and sex hormones can cause severe liver damage and cirrhosis and advise other transgender women against drinking [36, 37], so health concerns could also influence drinking behaviors among adolescents. Additional empirical evidence is needed to understand this disparity, and the influence of gender identity on decision to drink alcohol would be a good idea for future research. The study findings showed noticeable disparities between behavioral health of transgender and cisgender youths, and policy makers should consider adaptation and expansion of the "LGBTQIN + Well-Being Strategies in Thailand 2021–2023" of the nationally-influential Thai Health Promotion Foundation, particularly on how to expand the strategy-driven programs to include reduction of health disparities among secondary students [38]. Such efforts can be made in collaboration with the Foundation and partner organizations in the state and civil society sectors.

One-third of respondents who identified as transgender girls reported experiencing sexual violence within the past year. Furthermore, transgender boys and transgender girls were significantly more likely than cisgender girls to experience all types of violence in the past year. A study on Thai transgender women's experiences of stigma in daily lives showed various experiences of verbal, psychological and physical violence [12]. Although the Thai education has introduced sexuality education in its curriculum, such effort may not be effective in reducing homonegative attitudes among students [11]. The higher exposure to violence among transgender compared to cisgender respondents could be among the influences of the disparities in health behaviors. Policy makers should ensure that adequate violence prevention and support measures are in place for transgender youths in the school system, and that there are programs to address the narratives that influence the violence and discrimination. Such efforts may include but not be limited to expansion of local gender equality initiatives that aim to support both LGBTQI people and their significant others such as the 4P Project [39] and implementation of recommended policies on the Thai media's portrayal of LGBTQI people [40].

There were several issues with measurement questions in our study. In the depression measurement question, the phrase ("...*to the point where you could not do your routine daily function*...") was potentially problematic: there was no example of "routine daily function", and some participants may not have fully understood the question, which might have contributed to measurement errors. Future studies should consider incorporating a more standardized tool such as the Thai PHQ-2 [41], which does not contain any ambiguous wording. In addition, unsafe sexual behavior assessment questions used the term "having sex" without giving any details of what this act constituted. In addition to reactivity to the presence of classmates in their proximity, participants might also have varying definitions of "having sex", and might not have considered oral or anal intercourse to be 'sex', despite the potential exposure risk. A nationwide survey in Thailand showed that only half of the respondents considered male-male anal sex as "having sex", while one-fourth considered oral-genital contact as "having sex" [42]. Although the mentioned survey used a different reference word for "having sex", it is possible that such variation in definition of "having sex" also existed among our study participants. Future studies should include an introduction with a clearer definition of "having sex". Furthermore, the question on contraceptive use pertained only to birth control measures.

Transgender girls who had sex with members of the same sex (in anatomical terms) might have used condom for STD prevention but not for contraception and thus answered "No", while others might have understood the question to pertain to both contraception and STD prevention, and might have reported condom use that was actually intended for STD prevention. In that regard, the lower condom use among transgender boys who had sex with members of the same sex (in anatomical terms) was understandable. Future studies should include a question on STD prevention methods in addition to contraception in order to obtain a clearer and more complete measure of sexual behaviors. Drug use and unsafe sexual practices are sensitive behaviors whose reports can be influenced by social desirability. A previous study suggested that interviews assisted by electronic devices are non-inferior and potentially superior to the use of self-administered questionnaires in measuring sensitive behaviors [43]. We opted for paper-and-pencil questionnaire in this study because of logistical concerns, but future surveys should consider using tablet-assisted self-interviews or audio-computer-assisted self-interviews as potential alternatives.

### Strengths and limitations

The strengths of our study included the systematic classification of gender and the large sample size, which conveyed adequate statistical power to facilitate comparisons in our analyses. However, some limitations exist. First, although our survey did not include names or other identifying information, the survey was conducted in a classroom setting where proximity to other students may have given rise to the possibility of inadvertently disclosing confidential information to others. The data collection procedure, where an adult data collector came to the classroom and invited the students to participate, might not have allowed the students to feel adequately empowered to refuse to participate in the study thus compelling them to provide socially desirable answers. The fear of inadvertent disclosure and lack of empowerment to refuse to participate could have influenced the participants to provide socially desirable responses during their participation, and also presented an unforeseen ethical issue to be considered in future studies. Second, our outcomes did not include mental health outcomes that were common and significant in the health of transgender youths, such as gender dysphoria [22, 44]. Such questions should be considered for inclusion in future studies. Third, Thailand has a persistent problem of secondary school dropouts [45], thus the findings of this study could only be generalized to adolescents who remained in school.

### Conclusion

We found disparities in behavioral health including depression, suicidality, unsafe sexual behaviors, alcohol and tobacco consumption, drug use, and past-year experience of violence between Thai transgender and cisgender youths. However, we could only classify the participants into four broad categories (transgender boys, transgender girls, cisgender boys, and cisgender girls) and we did not measure sexual orientation of the participants. Future studies should further modify the measurement questions to expand the contribution of the findings to LGBTQ health. The findings nonetheless provide empirical data on the health of transgender youths, and highlighted the need for program managers and policy makers to consider existing local efforts to address health gaps in the LGBTQ community and expand such efforts to the school-going youths population.

### Supporting information

**S1 Dataset. Anonymized data set.** Anonymized data set to replicate the study findings. (CSV)

**S1 File. R codes.** Codes for data analyses, text file with annotations.
(TXT)

**S1 Checklist. STROBE checklist.** STROBE checklist for cross-sectional studies.
(DOCX)

## Acknowledgments

This study is a secondary data analysis. Primary data of the National School Survey on Alcohol Consumption, Substance Use and Other Health-Risk Behaviors 2016 were collected by the principal investigator and colleagues who allowed us to use the data.

## Author Contributions

**Conceptualization:** Wit Wichaidit, Virasakdi Chongsuvivatwong.

**Data curation:** Sawitri Assanangkornchai.

**Formal analysis:** Wit Wichaidit.

**Methodology:** Wit Wichaidit, Sawitri Assanangkornchai, Virasakdi Chongsuvivatwong.

**Supervision:** Sawitri Assanangkornchai, Virasakdi Chongsuvivatwong.

**Writing – original draft:** Wit Wichaidit.

**Writing – review & editing:** Wit Wichaidit, Sawitri Assanangkornchai, Virasakdi Chongsuvivatwong.

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
