## [Decision Letter · Decision Letter 0]

30 Dec 2020

PONE-D-20-21445

Disparities in Behavioral Health, Experience of Violence, and Problematic Parental Behaviors Between Cisgender and Transgender Thai Adolescents

PLOS ONE

Dear Dr. Wichaidit,

Thank you for submitting your manuscript to PLOS ONE. After careful consideration, we feel that it has merit but does not fully meet PLOS ONE’s publication criteria as it currently stands. Therefore, we invite you to submit a revised version of the manuscript that addresses the points raised during the review process.

We look forward to receiving your revised manuscript.

Kind regards,

Siyan Yi, MD, MHSc, PhD

Academic Editor

PLOS ONE

Journal Requirements:

2. Our internal editors feel that your manuscript is within the scope of our Health and Health Care in Gender Diverse Communities Call for Papers. The final collection of papers will be selected by a team of Guest Editors for PLOS ONE: Asa Radix (New York University/Langone Health), Jae Sevelius (University of California San Francisco), and Ayden Scheim (Drexel University). Additional information can be found on our announcement page: https://collections.plos.org/s/health-gender-diversity. If you would like your manuscript to be considered for this collection, please let us know in your cover letter and we will ensure that your paper is treated as if you were responding to this call. If you would prefer to remove your manuscript from collection consideration, please specify this in the cover letter.

3.We note that you have indicated that data from this study are available upon request. PLOS only allows data to be available upon request if there are legal or ethical restrictions on sharing data publicly. For information on unacceptable data access restrictions, please see http://journals.plos.org/plosone/s/data-availability#loc-unacceptable-data-access-restrictions.

Reviewers' comments:

Reviewer's Responses to Questions

**Comments to the Author**

1. Is the manuscript technically sound, and do the data support the conclusions?

Reviewer #1: Partly

Reviewer #2: No

Reviewer #3: Yes

2. Has the statistical analysis been performed appropriately and rigorously? 

Reviewer #1: Yes

Reviewer #2: Yes

Reviewer #3: Yes

3. Have the authors made all data underlying the findings in their manuscript fully available?

Reviewer #1: Yes

Reviewer #2: Yes

Reviewer #3: Yes

4. Is the manuscript presented in an intelligible fashion and written in standard English?

Reviewer #1: Yes

Reviewer #2: Yes

Reviewer #3: Yes

5. Review Comments to the Author

Reviewer #1: This paper sought to deal with an important topic, the disparities in health and behavioral outcomes between cisgender and transgender adolescents in Thailand. However, this paper has several critical weaknesses.

*** Major comments

Comment 1.

A notable strength of this study is that it compares a wide variety of health outcomes, health-related behaviors, and experiences between cisgender and transgender Thai adolescents using a large, nationwide sample of school-going Thai adolescents. The study’s outcomes could be categorized into three groups.

Group 1: mental health and health behaviors

Group 2: experiences of violence

Group 3: parental addictive and violent behaviors

Currently, the results for the three groups are presented together in table 3. Analyses regarding these variables should be revisited to take into consideration for their differing implications.

First of all, it is strongly advised that the author drops variables in group 3 – parental addictive and violent behaviors – for a couple of reasons. Group 3 variables are different from group 1 and 2 variables in that they are not about the participants themselves, but about their parents. More importantly, the conclusions drawn on parental behaviors by the author (lines 400-407, page 31) can potentially be interpreted to blame transgender youths for their parents’ addictive and violent behaviors. For these reasons, the reviewer strongly recommends the author to drop group 3 variables from the study. This could be a research topic for another paper.

Second, the reviewer recommends that the results for groups 1 and 2 are presented in separate tables. Variables in group 1 are indicators of participants’ health, and variables in group 2 are about experiences of violence. It is highly possible that the experience of group 2 could be used to explain the observed health disparities of the group 1 results. It is suggested that the author first presents the results from group 1 variables and describes the health disparities between cisgender and transgender Thai adolescents. This should then be followed by results from group 2 variables in a separate table, with the discussion of these results in the context of health disparities shown in group 1 results.

Comment 2.

Although the results clearly show the health disparities experienced by transgender youths in Thailand, this paper did not provide sufficient information about the hostile social environment against transgender youth in Thailand. The author does briefly mention the negative social situations transgender youths in Thai face in the introduction (lines 57-67, pages 4-5). It is suggested that the author includes specific examples of unfair social and institutional conditions Thai transgender youths experience to better contextualize the main results.

Comment 3.

In the paper, assigned female at birth (AFAB) and assigned male at birth (AMAB) are categories of participants who did not provide an answer for the question on their gender identity. According to the paragraph that starts in line 353 of page 29, author also mentioned that they might be the people who did not understand why a separate question was asked on gender or the people who do not want reveal their gender identity, and people with missing responses. So it is evident that authors do not have enough information to understand who AFAB and AMAB actually are. Therefore, it is impossible to discuss about their findings. It is strongly recommended that the author excludes those with missing gender information from the analysis.

Comment 4.

This paper compares health statuses of Thai transgender adolescents only against their cisgender male counterparts (reference group). There is no rationale as to why cisgender male adolescents were chosen as the sole reference group to investigate health statuses of adolescents of different gender identities. Please provide a rationale as to why the author chose cisgender male adolescents as the reference group. In the events that the author cannot provide the rationale, it is suggested that they review previous works that have compared cisgender and transgender health outcomes (Downing & Przedworski, 2018; Lee et al., 2020; Thoma et al., 2019) and revise the analyses accordingly.

Downing, J. M., & Przedworski, J. M. (2018). Health of transgender adults in the US, 2014–2016. American journal of preventive medicine, 55(3), 336-344.

Lee, H., Operario, D., van den Berg, J. J., Yi, H., Choo, S., & Kim, S. S. (2020). Health disparities among transgender adults in South Korea. Asia Pacific Journal of Public Health, 32(2-3), 103-110.

Thoma, B. C., Salk, R. H., Choukas-Bradley, S., Goldstein, T. R., Levine, M. D., & Marshal, M. P. (2019). Suicidality disparities between transgender and cisgender adolescents. Pediatrics, 144(5).

Comment 5A.

The paragraph that starts at line 368 in page 30 should be revised. The paragraph is not well-organized and contains statements that are concerning. First, the author makes unnecessary comparisons with other countries (e.g. US). Since the study is not about making international comparisons of transgender youth health status, these comparisons seem unnecessary. Furthermore, the comparison sentence colludes with the findings. Table 3 shows that transgender female and transgender male adolescents have 1.82 times and 2.06 times higher odds of attempting suicide compared to cisgender male adolescents. This result, along with results of other indicators of mental health, suggests that the mental health disparities between cisgender and transgender adolescents in Thailand are severe. Despite the severity of health disparities, the author writes that suicidality in transgender adolescents is lower in Thailand when compared to other countries only to highlight the culture of acceptance of transgender identity in Thailand. The reviewer is concerned that these sentences that make comparisons may downplay the apparent severity of the mental health gap experienced by transgender adolescents in Thailand.

Comment 5B.

In the same paragraph from the comment above, the author seeks to explain the differences in mental health outcomes between transgender male and transgender female adolescents without evidence and proper citation. If authors want to discuss differences between the two populations, authors should carry out statistical analyses that directly compares the health of transgender male and transgender female adolescents to assess whether there are statistically significant differences between the two groups.

***Minor comments

Comment 1.

Table 2 shows the distribution of basic characteristics among study respondents by their gender identities. In addition to representing the distribution of the characteristics, I suggest that the author includes results of chi-square tests, to show any significantly different distributions of these characteristics among participants of different gender identities.

Comment 2.

Past research on health disparities between cisgender and transgender individuals have included measures of income or social status in their lists of covariates (Downing & Przedworski, 2018; Lee et al., 2020; Thoma et al., 2019). It is suggested that the author includes an indicator of socioeconomic status such as household income (Downing & Przedworski, 2018; Lee et al., 2020) or perceived social status (Thoma et al., 2019) as another covariate.

Downing, J. M., & Przedworski, J. M. (2018). Health of transgender adults in the US, 2014–2016. American journal of preventive medicine, 55(3), 336-344.

Lee, H., Operario, D., van den Berg, J. J., Yi, H., Choo, S., & Kim, S. S. (2020). Health disparities among transgender adults in South Korea. Asia Pacific Journal of Public Health, 32(2-3), 103-110.

Thoma, B. C., Salk, R. H., Choukas-Bradley, S., Goldstein, T. R., Levine, M. D., & Marshal, M. P. (2019). Suicidality disparities between transgender and cisgender adolescents. Pediatrics, 144(5).

Comment 3.

Depending on the outcome variable, there were varying numbers of participants dropped due to missing values. This leads to each regression having different sample sizes. Please include sample sizes, frequencies, and the proportions (%) in each of the relevant cells in tables 1, 2, and 3 to clearly provide the results with missing data.

Comment 4.

In the following sentence, “We then performed multivariate analyses to measure the association between gender classification and the outcomes adjusting for the effect of basic characteristics of the respondents using multivariate logistic regression (lines 244-246, page 14),” please include a phrase or sentence stating that the multivariate analyses were performed separately for each of the outcome variables.

Comment 5.

Please check the consistency of variable names the author used throughout the manuscript. For instance, the author used the word ‘suicidal ideation’ in the abstract and in other parts, ‘suicide contemplation’ in the method part, and ‘considered suicide’ in table 3. And in checking for the consistency, please refer to prior research for most commonly used terminology regarding your outcomes. For example, regarding suicidal behaviors, the most commonly used terminologies include ‘suicidal ideation’ and ‘suicide attempt.’

Comment 6.

There are various health outcomes and health behaviors that have high prevalence, such as 51.1% (ever drinker among transgender male adolescents) and 22.6% (prevalence of depressive experiences among transgender male adolescents). However, the author has used logistic regression for all of their analyses and reports adjusted odds ratios, which would lead to overestimation of the associations. It is suggested that the author runs a modified Poisson regression instead of logistic regression, according to the work of Zou (2004).

Zou, G. (2004). A modified Poisson regression approach to prospective studies with binary data. American Journal of Epidemiology, 159(7), 702-706.

Comment 7.

In Table 2 & 3, The N’s in the second row total out to 38,186, which is different from the n given in the title (38,189). Please check the numbers.

Comment 8.

In United States Center for Disease Control and Prevention, the overall definition of Intimate partner violence is “physical violence, sexual violence, stalking and psychological aggression (including coercive tactics) by current or former intimate partner (i.e., spouse, boyfriend/girlfriend, dating partner, or ongoing sexual partner).” (Breiding, 2015, p. 11)

The perpetrators of IPV defined in your study are “significant other”, which may include family, relative, friend, or acquaintance as well as partner or spouse so on. Therefore, It is recommended to change the term to “Interpersonal violence” rather than IPV, which is used as an idiom.

Breiding, M., Basile, K. C., Smith, S. G., Black, M. C., & Mahendra, R. R. (2015). Intimate partner violence surveillance: Uniform definitions and recommended data elements. Version 2.0.

Comment 9.

Please match intext citation styles.

Comment 10.

The study shows that Thai transgender adolescents experience severe disparities in health compared to their cisgender counterparts. Please provide policy implications and/or specific programs to address this public health issue. It is also helpful to provide examples of policies and programs.

Comment 11.

There is no discussion on the disparities in experiences of violence between cisgender and transgender adolescents in Thailand. Please provide discussions according to the findings.

Comment 12.

The reasons for including ever having had sexual experience and having 4 or more partners as indicators of health-related behavior are unclear. Please provide necessary rationales for including these outcomes or consider dropping them from the analysis.

Comment 13.

Given that “male’’ and “female” are adjectives, please denote the noun that ‘male’ and ‘female’ are explaining (e.g. transgender males is wrong; it should be transgender male adolescents in this context).

Reviewer #2: This paper covers an important topic, but does it in a clumsy and incomplete manner. It fails to sufficiently engage with the literature about disparities between transgender and cisgender youth that exists in Thailand. The paper's introduction needs to be improved, its methods/procedures section needs to be shortened, the data presented in tables needs to be culled and data that is not used in the results section discussion or overall discussion needs to be removed. The use of 6 categories of youth is confusing and what AMAB and AFAB are is left to the imagination of the reader.

I suggest a very major revision.

Specific comments:

Line 39: I think the paragraph should start with an explanation of the word 'gender spectrum', i.e. that gender can be seen as a fluid, multi-dimensional concept, not as a binary term.

Line 45: sexual behaviors are not necessarily detrimental to the health of adolescents and it looks strange to see them listed on par with drug use and suicidal behaviors. I suggest to add the word 'unsafe' before 'sexual behaviors', to refer to unprotected sexual behaviors that can lead to negative outcomes such as HIV, STI or unwanted pregnancy.

Line 48: there is evidence from Thailand for this statement. See Van Griensven F, Kilmarx PH, Jeeyapant S, Manopaiboon C, Korattana S, Jenkins RA, Uthaivoravit W, Limpakarnjanarat K, Mastro TD. The prevalence of bisexual and homosexual orientation and related health risks among adolescents in northern Thailand. Archives of sexual behavior. 2004 Apr 1;33(2):137-47.

Line 49-50: there is evidence from Thailand for this statement: Guadamuz TE, Wimonsate W, Varangrat A, Phanuphak P, Jommaroeng R, McNicholl JM, Mock PA, Tappero JW, van Griensven F. HIV prevalence, risk behavior, hormone use and surgical history among transgender persons in Thailand. AIDS and Behavior. 2011 Apr 1;15(3):650-8.

and

Yadegarfard M, Ho R, Bahramabadian F. Influences on loneliness, depression, sexual-risk behaviour and suicidal ideation among Thai transgender youth. Culture, health & sexuality. 2013 Jun 1;15(6):726-37.

and

Boonchooduang N, Louthrenoo O, Likhitweerawong N, Charnsil C, Narkpongphun A. Emotional and behavioral problems among sexual minority youth in Thailand. Asian journal of psychiatry. 2019 Oct 1;45:83-7.

and

Kittiteerasack P, Matthews AK, Steffen A, Corte C, McCreary LL, Bostwick W, Park C, Johnson TP. The influence of minority stress on indicators of suicidality among lesbian, gay, bisexual and transgender adults in Thailand. Journal of Psychiatric and Mental Health Nursing. 2020 Nov 15.

and

Logie CH, Newman PA, Weaver J, Roungkraphon S, Tepjan S. HIV-related stigma and HIV prevention uptake among young men who have sex with men and transgender women in Thailand. AIDS patient care and STDs. 2016 Feb 1;30(2):92-100.

There are a few more papers on transgender and sexual minority youth from Thailand that should be reviewed / mentioned in this section, and I would suggest there is no need to include evidence from transgender people in Canada or USA, as the social and cultural context there is completely different.

Line 50-55: See above, it is not true that there is no Thai data on disparities between transgender and cisgender youth.

Line 57: Thailand is not uniformly tolerant to transgender people. Please review these papers:

Matzner A. The complexities of acceptance: Thai student attitudes towards kathoey. Crossroads: An Interdisciplinary Journal of Southeast Asian Studies. 2001 Jan 1:71-93.

Ojanen TT, Newman PA, Ratanashevorn R, de Lind van Wijngaarden JW, Tepjan S. Whose paradise? An intersectional perspective on mental health and gender/sexual diversity in Thailand.

de Lind van Wijngaarden JW, Fongkaew K. “Being Born like This, I Have No Right to Make Anybody Listen to Me”: Understanding Different Forms of Stigma among Thai Transgender Women Living with HIV in Thailand. Journal of Homosexuality. 2020 Aug 26:1-8.

Manalastas EJ, Ojanen TT, Torre BA, Ratanashevorn R, Hong BC, Kumaresan V, Veeramuthu V. Homonegativity in southeast Asia: Attitudes toward lesbians and gay men in Indonesia, Malaysia, the Philippines, Singapore, Thailand, and Vietnam. Asia-Pacific Social Science Review. 2017 Jun 1;17(1):25-33.

Line 57-61: I suggest to rewrite this, there are several surveys and studies about Thai Transgender women, but it is true that far less is known about transgender men.

Page 6 in the document is empty, suggest to remove.

Page 8-12: I suggest to remove all this information or make it shorter. There is no need to introduce each and every question asked and explain why these questions were included.

The section on Procedure should be shortened or deleted.

Line 222-230: Important to include a limitation on the methodology used. Important but little known evidence from Thailand has shown very large differences in outcomes depending on which data collection method was used. Self-administered written questionnaires resulted in lower self-reporting of 'shameful' behaviors than when a palm-top / iPad device was used for data collection. Please see: Van Griensven F, Naorat S, Kilmarx PH, Jeeyapant S, Manopaiboon C, Chaikummao S, Jenkins RA, Uthaivoravit W, Wasinrapee P, Mock PA, Tappero JW. Palmtop-assisted self-interviewing for the collection of sensitive behavioral data: randomized trial with drug use urine testing. American journal of epidemiology. 2006 Feb 1;163(3):271-8.

Line 262 - beginning of Results:

The AMAB / AFAB categories sound a bit silly, if I may say so. More explanation is needed here. Does it mean these boys and girls are gay/lesbian/bisexual/uncertain about their sexuality? It is useful to explain in the paper somewhere that Thai (at least common Thai) does not have separate words for gender and sexuality, both are referred to as 'phet'. This makes it difficult to interpret/translate. Is it possible to simplify the paper, for example by focusing only on transgender students versus cisgender students? Or is it possible to combine all cisgender students and compare them with all transgender students regardless of gender? Or include AMAB/AFAB as 1 category, transgender men&women as 1 category, and all cisgender youth as 1 category? It feels overwhelming to have so many cells and so many categories, and it becomes hard to see what the focus is and what the result of the analysis suggests. I think more work and discussion needs to go into this analysis, and maybe the focus of the paper should be better grounded in previously conducted research on this issue.

Table 3 is way too long, and contains too much information. It is almost as if the authors included the results of the questionnaire and ask the reader to do their own analysis. I suggest to be more selective in presenting information, and break the table up into several smaller tables, and maybe merge some of the sub-populations.

Line 308-310: Where does the focus on parental problems come from? Do the authors think that parents are to blame for their child being transgender? Or do they think the transgender identity of the child leads parents to have problems? I do not understand the link. Out the enormous amount of data in Table three, only a few small points are discussed in this section, and this is one of them. I do not think this section is solid enough.

Line 327-330: This suggests that AMAB may in fact be same-sex attracted youth? Apparently 'party drug' use ('ice') is high among young gay men. There are a few papers on this to review using Thai data.

Line 357: That is nonsense, using 1 study to claim that the percentage of non-normative gender identities in any given population has to be <1%. See the papers about vocational students in Northern Thailand and several other papers from Thailand that have assessed the prevalence of homosexual/bisexual orientation and transgender identities, and they are all much higher than 1%.

Line 367: This is an interesting finding. In comparison, see this paper: https://www.sciencedirect.com/science/article/abs/pii/S0049089X16308468

Line 370-376: When discussing Thai media reporting on LGBTQ people, I suggest to review and reference this paper:

Fongkaew K, Khruataeng A, Unsathit S, Khamphiirathasana M, Jongwisan N, Arlunaek O, Byrne J. “Gay Guys are Shit-Lovers” and “Lesbians are Obsessed With Fingers”: The (Mis) Representation of LGBTIQ People in Thai News Media. Journal of homosexuality. 2019 Jan 28;66(2):260-73.

Line 408-418 should move to the limitations section below.

413-418: Very good point.

Conclusion is very short. If this is what the paper is about, it will be easy to slim it down, removing all other information presented in tables but not discussed.

Reviewer #3: Overall, I think that the large sample size, robust sampling strategy, and wide selection of variables makes this study a very worthy contribution to the literature on transgender mental health in Thailand. Below, I make some observations and suggestions that I hope will be helpful to contextualizing the findings better.

Terminology: The terms "transgender male" and "transgender female" are a bit confusing. "Male" and "female" are often used to refer to biological sex rather than self-ascribed gender; having read the abstract, I am still unsure whether "transgender male" and "transgender female" refer to individuals of male or female birth sex, and consider themselves boys or girls. Wonder if the abstract word limit makes it possible to clarify the terms in brief there as well? Also, given that the entire study was on adolescents, it might make sense using "transgender boys," "transgender girls" and "cisgender boys," "transgender girls" respectively (the terms "boys" and "girls" might capture the self-ascribed gender better than the strictly biological terms "male" or "female").

lines (l.) 59 - 67: There are quite a few studies conducted in Thailand especially on violence and depression that you might want to take into account here - they may not use exactly the same kind of disaggregation as the present study, but are nevertheless relevant. In the light of these studies, in particular the statement (l. 59-60) that existing studies are based on small-scale surveys or qualitative data is not exactly accurate (e.g., https://doi.org/10.1080/13691058.2020.1737235 ; https://doi.org/10.1371/journal.pone.0237707 ; https://doi.org/10.1080/10826084.2019.1638936 ; http://unesdoc.unesco.org/images/0022/002275/227518e.pdf ; https://so03.tci-thaijo.org/index.php/jpss/article/view/102396) ; Table 5 in the Plan International Thailand report on school bullying that relies on self-reported level of masculinity uses a proxy variable for disaggregating transgender and gender-nonconforming youth from those who are gender conforming.

l. 130 Wonder if the Thai terminology used for "sexual intercourse" made it clear to participants what kind of sex was being asked about? And did the participants understand it in the same way as the researchers? There is some older evidence that many Thai people don't count oral or anal contact as "intercourse" - see p. 33 in the following article: https://doi.org/10.1300/J041v09n02_02 - it might be good to reflect on whether this poses any limitations to the validity of the analyses? I see that later in the article, the Limitations section notes this, but the above article may help to contextualize this concern.

l. 142 It may be good to note the limits of a question on contraception in, for example, male-male or female-female sexual acts (is it understood by 1-the survey makers and 2-adolescents to refer to birth control only, or to preventing STDs and HIV as well?). This links to the lowered odds ratio of "trans males" using condoms according to Table 3 - most of them are likely to have had sex with females, so it's (in anatomical terms) female-female sex, where condom use is understandably less common than in male-female or male-male sex).

Table 1. The high percentage of participants not answering the "gender identity" question might be a result of the participants' confusion in being asked about their identity and yet being given only two response options (because an identity question corresponding to the Thai understanding of self-defined phet might rather give a long list of locally recognized identities, such as man, woman, kathoey, gay, tom, dee, etc. where gender identity and sexual orientation are aspects of the same categorical folk identity construct "phet"- see Table 1 in the Plan International report noted above for an example; my guess would be that those identifying as gay, bi, dee, etc. might be quite prevalent among those not answering the second question on their self-ascribed phet). ... this also provides clues to locally relevant response options to the survey as discussed in lines 338-342. It also calls into question whether the statement in lines 347-349 about homosexual and heterosexual persons of the same gender identity being grouped together is correct, and provides an alternative explanation to the statement in lines 356-360.

Table 2. Wonder if providing by-row rather than by-column percentages would be more useful to the reader? This would make it easier to compare the prevalence of each gender/sex group across types of demographic groups.

l. 364 The statement of there being little discrimination from teachers is questionable. For example, a World Bank commissioned survey ( http://documents1.worldbank.org/curated/en/319291524720667423/pdf/124554-v2-main-report-Economic-Inclusion-of-LGBTI-Groups-in-Thailand-Report-Thai-Version-PUBLIC.pdf ) found that 23% of transgender adults reported having been discriminated in education (p. 40). However, there may be a difference in discrimination from teachers as individuals versus educational institutes as institutions.

l. 368-389 The suicidality figures and their possible antecedents should be compared with some of the studies on Thai adolescents I mentioned above. There are also other studies that may be relevant, such as emerging Thai studies on LGBTIQ adults (e.g., https://doi.org/10.1111/jpm.12713 ) or a couple of earlier articles on Thai youth using convenience samples (https://doi.org/10.1080/19361653.2014.910483 ; http://dx.doi.org/10.1080/13691058.2013.784362 ).

l. 404 Footnote style citation missing for Katz-Wise et al

6. PLOS authors have the option to publish the peer review history of their article (what does this mean?). If published, this will include your full peer review and any attached files.

Reviewer #1: No

Reviewer #2: No

Reviewer #3: **Yes: **Timo Tapani Ojanen

---

## [Author Response · Author response to Decision Letter 0]

26 Feb 2021

Dear Reviewers,

My co-authors and I would like to thank you for the constructive and thoughtful comments. We have responded to them on a point-by-point basis in the "Response to Reviewers.docx" document. We thank you for your kind consideration and we look forward to hearing from you.

Sincerely Yours,

Corresponding Author

---

## [Decision Letter · Decision Letter 1]

7 Apr 2021

PONE-D-20-21445R1

Disparities in Behavioral Health and Experience of Violence between Cisgender and Transgender Thai Adolescents

PLOS ONE

Dear Dr. Wichaidit,

Thank you for submitting your manuscript to PLOS ONE. After careful consideration, we feel that it has merit but does not fully meet PLOS ONE’s publication criteria as it currently stands. Therefore, we invite you to submit a revised version of the manuscript that addresses the points raised during the review process.

We look forward to receiving your revised manuscript.

Kind regards,

Siyan Yi, MD, MHSc, PhD

Academic Editor

PLOS ONE

Journal Requirements:

Reviewers' comments:

Reviewer's Responses to Questions

**Comments to the Author**

1. If the authors have adequately addressed your comments raised in a previous round of review and you feel that this manuscript is now acceptable for publication, you may indicate that here to bypass the “Comments to the Author” section, enter your conflict of interest statement in the “Confidential to Editor” section, and submit your "Accept" recommendation.

Reviewer #2: All comments have been addressed

Reviewer #3: (No Response)

2. Is the manuscript technically sound, and do the data support the conclusions?

Reviewer #2: Yes

Reviewer #3: Yes

3. Has the statistical analysis been performed appropriately and rigorously? 

Reviewer #2: Yes

Reviewer #3: Yes

4. Have the authors made all data underlying the findings in their manuscript fully available?

Reviewer #2: Yes

Reviewer #3: Yes

5. Is the manuscript presented in an intelligible fashion and written in standard English?

Reviewer #2: No

Reviewer #3: Yes

6. Review Comments to the Author

Reviewer #2: Dear authors

I think you have done a stellar job addressing my (and other reviewers') comments.

I have only two minor points. While I am glad you clarified the issue of acceptance/tolerance of transgender people in Thai society somewhat, I do not think the level of 'homonegativity' in Thailand is higher than in neighboring countries, that would take the statement too far! It is a pity you could not access the Matzner paper I sugggested, it is really fascinating and would provide a theoretical explanation for the contradictions that characterise this issue in Thai society; maybe a colleague with access to paid content can find it for you?

Also when describing that half of respondents would consider male-male sex to be 'sex', I would add the word 'only'...

I.e. only half of the participants.... This is an important and striking finding :)

I like it that you have include linguistic challenges regarding gender/sexuality and agree that sexual orientation is another complicating factor.

In general, I suggest a final edit by a native English speaker, and then I think the paper should be ready for publication.

Best wishes

Jan ยันต์

Reviewer #3: I feel the article is overall in much better shape now, following the authors' extensive edits to the manuscript. My comments below are mostly minor language edits and corrections to wordings the authors may have forgotten to adjust following overall changes to the manuscript. Once these have been fixed, I think the paper will be ready for publication.

Minor comments:

Abstract:

- "identified as transgenders" -> "identified as transgender"

- "higher prevalence suicidal ideation" -> "had a higher prevalence of suicidal ideation"

- "school-going youths population" -> "the school-going youth population"

Introduction:

- watch out for unnecessary use of "still" (e.g., l. 58, 59)

l. 56: Would be helpful to clarify what kinds of risk behavior. Sexual? Substances? Extreme sports?

l. 58-59: "still has higher prevalence of homonegativity..." To represent Manalastas et al. more accurately, you could state that the level of homonegative attitudes in Thailand is higher than in some other countries in the region, and lower than in others (better still, state which SE Asian countries included in the study are more and which countries are less homonegative than Thailand)

l. 60 "despite perception as a safe haven" -> "although many perceive it as a safe haven"

l 61 "transgender youths are" -> "transgender youth"

l. 68-69 "as well as parental issues related to addiction and violence," -> remove if no longer included in analyses

l. 72 "sex assigned at birth (sex)" -> "sex assigned at birth"

l. 74 "and parental problematic behaviors" -> remove if no longer included in analyses

l. 116 Does "withdrawal" here refer to the withdrawal method of contraception (coitus interruptus)? Please clarify if the original survey item implied this.

l. 141 "Sexual violence" -> "sexual violence"

l. 167-168 "and parental addictive and violent behaviors)" -> remove if no longer included

l. 196-197 "Transgender females and transgender males" -> adjust in keeping with revised terminology

l. 223 "had significantly prevalence of lifetime" -> "had significantly higher prevalence of lifetime"

l. 259 "What gender do you identify as?" - in this case, which Thai words would you use to translate "gender" and "identify" so as to understandable to school-age kids? If you have a suggestion for these, you might include the transcription of your preferred Thai words for these terms in brackets after the word; this would be useful to other researchers

l. 263 Considering current developments of gender identity in Thailand, including "non-binary" as a response option would also make sense

l. 290 "minority stress" (you could give a brief definition for it)

l. 294-295 change terminology to match the rest of the article (transgender males->transgender boys; cisgender males->cisgender boys; transgender females->transgender girls)

l. 297 "transgender students" -> "transgender boys" (their aspiration to masculinity is the key point here)

l. 316 "These higher" -> "The higher"

l. 322 "the 4P Project" - instead of the cited popular-press article, please cite the project report, which has now been completed and is available at https://lsed.tu.ac.th/uploads/lsed/pdf/research/%20LGBT4P.pdf

7. PLOS authors have the option to publish the peer review history of their article (what does this mean?). If published, this will include your full peer review and any attached files.

Reviewer #2: No

Reviewer #3: **Yes: **Timo Tapani Ojanen

---

## [Author Response · Author response to Decision Letter 1]

27 Apr 2021

Response to Reviewers

PONE-D-20-21445R1

Disparities in Behavioral Health and Experience of Violence between Cisgender and Transgender Thai Adolescents

Reviewers' comments: 

Reviewer #2: 

COMMENT: I think you have done a stellar job addressing my (and other reviewers') comments.

RESPONSE: Thank you so much

COMMENT: I have only two minor points. While I am glad you clarified the issue of acceptance/tolerance of transgender people in Thai society somewhat, I do not think the level of 'homonegativity' in Thailand is higher than in neighboring countries, that would take the statement too far! It is a pity you could not access the Matzner paper I sugggested, it is really fascinating and would provide a theoretical explanation for the contradictions that characterise this issue in Thai society; maybe a colleague with access to paid content can find it for you?

RESPONSE: We thank the reviewer for the suggestion. We decided to revise the opening sentence in the third paragraph of the INTRODUCTION section to the following: 

“Thailand is a middle-income country in South East Asia that experiences homonegativity, not unlike other countries in the region[10].”

COMMENT: Also when describing that half of respondents would consider male-male sex to be 'sex', I would add the word 'only'...

I.e. only half of the participants.... This is an important and striking finding :)

RESPONSE: Agree. The sentence has been revised as follows: 

“A nationwide survey in Thailand showed that only half of the respondents considered male-male anal sex as "having sex", while one-fourth considered oral-genital contact as "having sex"[42]”

COMMENT: I like it that you have include linguistic challenges regarding gender/sexuality and agree that sexual orientation is another complicating factor. 

RESPONSE: Thank you

COMMENT: In general, I suggest a final edit by a native English speaker, and then I think the paper should be ready for publication.

Best wishes

Jan ยันต์ 

RESPONSE: Thank you very much for the compliments and constructive comments ขอบคุณครับ

Reviewer #3 

COMMENT: I feel the article is overall in much better shape now, following the authors' extensive edits to the manuscript. My comments below are mostly minor language edits and corrections to wordings the authors may have forgotten to adjust following overall changes to the manuscript. Once these have been fixed, I think the paper will be ready for publication. 

RESPONSE: Thank you. The other reviewer, a native speaker of English, has graciously provided detailed instructions for language edits, which we have followed and made changes throughout the manuscript.

Minor comments: 

Abstract: 

COMMENT: 

- "identified as transgenders" -> "identified as transgender"

- "higher prevalence suicidal ideation" -> "had a higher prevalence of suicidal ideation"

- "school-going youths population" -> "the school-going youth population"

RESPONSE: Thank you for the suggestions. We have made changes accordingly. 

Introduction: 

COMMENT: - watch out for unnecessary use of "still" (e.g., l. 58, 59)

RESPONSE: The first two sentences of the third paragraph of the INTRODUCTION section has been revised as follows: 

“Thailand is a middle-income country in South East Asia that experiences homonegativity, not unlike other countries in the region[10]. Although many perceive the country as a safe haven[11], Thai transgender women do report experiences of discrimination and violence[12].”

COMMENT: l. 56: Would be helpful to clarify what kinds of risk behavior. Sexual? Substances? Extreme sports? 

RESPONSE: To avoid redundancy in mentioning citation [5] twice, we changed the middle part of the second paragraph to the following: 

“Behavioral health issues affect both transgender and cisgender adolescents, although a previous study has shown disparities in the prevalence of unsafe sexual practices, sexual violence, drug use, and depression [5].”

COMMENT: l. 58-59: "still has higher prevalence of homonegativity..." To represent Manalastas et al. more accurately, you could state that the level of homonegative attitudes in Thailand is higher than in some other countries in the region, and lower than in others (better still, state which SE Asian countries included in the study are more and which countries are less homonegative than Thailand)

RESPONSE: We thank the reviewer for the comment. The other reviewer has also commented on this sentence and remarked that they “do not think the level of 'homonegativity' in Thailand is higher than in neighboring countries, that would take the statement too far!...”

Therefore, as a compromise, we decided to revise the first two sentences of the third paragraph of the INTRODUCTION section to the following: 

“Thailand is a middle-income country in South East Asia that experiences homonegativity, not unlike other countries in the region[10]. Although many perceive the country as a safe haven[11], Thai transgender women do report experiences of discrimination and violence[12].”

COMMENT: l. 60 "despite perception as a safe haven" -> "although many perceive it as a safe haven"

RESPONSE: Change made

COMMENT: l 61 "transgender youths are" -> "transgender youth"

RESPONSE: Sentence revised as follows: 

“Although many perceive the country as a safe haven[11], Thai transgender women do report experiences of discrimination and violence[12].”

COMMENT: l. 68-69 "as well as parental issues related to addiction and violence," -> remove if no longer included in analyses

RESPONSE: Removal made

COMMENT: l. 72 "sex assigned at birth (sex)" -> "sex assigned at birth" 

RESPONSE: Change made

COMMENT: l. 74 "and parental problematic behaviors" -> remove if no longer included in analyses

RESPONSE: Removal made

COMMENT: l. 116 Does "withdrawal" here refer to the withdrawal method of contraception (coitus interruptus)? Please clarify if the original survey item implied this.

RESPONSE: Yes. The most direct translation of the question item would be “external ejaculation” (“หลั่งภายนอก”), which (I believe) corresponds to coitus interruptus. We have revised the term to the following:

“Withdrawal (coitus interruptus)”

COMMENT: l. 141 "Sexual violence" -> "sexual violence"

RESPONSE: Change made

COMMENT: l. 167-168 "and parental addictive and violent behaviors)" -> remove if no longer included

RESPONSE: Change made

COMMENT: l. 196-197 "Transgender females and transgender males" -> adjust in keeping with revised terminology

RESPONSE: Changed to “Transgender girls and transgender boys”

COMMENT: l. 223 "had significantly prevalence of lifetime" -> "had significantly higher prevalence of lifetime"

RESPONSE: Change made

COMMENT: l. 259 "What gender do you identify as?" - in this case, which Thai words would you use to translate "gender" and "identify" so as to understandable to school-age kids? If you have a suggestion for these, you might include the transcription of your preferred Thai words for these terms in brackets after the word; this would be useful to other researchers

RESPONSE: That’s an interesting question. We feel that the word "ตัวตน" would be an appropriate translation of "identity" for the target population. Actually, we changed the wording in the 2020 student health survey questionnaire from "เพศวิถี (คุณคิดว่าคุณเป็นเพศ...)" to "เพศวิถี (ตัวตนคุณที่แท้จริงเป็นเพศใด)". During the pilot-testing of the questionnaire, no student expressed confusion with regard to the question wording. That said, we feel that such suggestion using the word ตัวตน alone without its context would not be as useful to future researchers. Thus we have included the transcription of the original and preferred Thai words in their sentences in the revised manuscript as follows:

“Future studies should consider modification of the gender identity measurement question to help reduce this non-response, e.g., changing from "You think you are..." ("Khun kid waa khun ben phet...") to "What gender do you identify as?" ("Tuaton tii tae jing khun ben phet dai") to reflect the notion that gender identity is firmly felt and integral to one's being”

COMMENT: l. 263 Considering current developments of gender identity in Thailand, including "non-binary" as a response option would also make sense

RESPONSE: Change made

COMMENT: l. 290 "minority stress" (you could give a brief definition for it)

RESPONSE: Sentenced revised as follows: 

“A previous study found minority stress (i.e., stress faced by members of stigmatized minority groups caused by factors such as lack of social support, low socioeconomic status, interpersonal prejudice, and discrimination) in the Thai homosexual and bisexual men population[33].”

COMMENT: l. 294-295 change terminology to match the rest of the article (transgender males->transgender boys; cisgender males->cisgender boys; transgender females->transgender girls)

RESPONSE: Changes made

COMMENT: l. 297 "transgender students" -> "transgender boys" (their aspiration to masculinity is the key point here)

RESPONSE: Change made

COMMENT: l. 316 "These higher" -> "The higher"

RESPONSE: Change made

COMMENT: l. 322 "the 4P Project" - instead of the cited popular-press article, please cite the project report, which has now been completed and is available at https://lsed.tu.ac.th/uploads/lsed/pdf/research/%20LGBT4P.pdf

RESPONSE: We have added the new reference to replace the existing one

---

## [Editor Report · Decision Letter 2]

18 May 2021

Disparities in Behavioral Health and Experience of Violence between Cisgender and Transgender Thai Adolescents

PONE-D-20-21445R2

Dear Dr. Wichaidit,

We’re pleased to inform you that your manuscript has been judged scientifically suitable for publication and will be formally accepted for publication once it meets all outstanding technical requirements.

Kind regards,

Siyan Yi, MD, MHSc, PhD

Academic Editor

PLOS ONE
---

## [Editor Report · Acceptance letter]

20 May 2021

PONE-D-20-21445R2 

Disparities in Behavioral Health and Experience of Violence between Cisgender and Transgender Thai Adolescents 

Dear Dr. Wichaidit:

I'm pleased to inform you that your manuscript has been deemed suitable for publication in PLOS ONE. Congratulations! Your manuscript is now with our production department. 

Kind regards, 

on behalf of

Dr. Siyan Yi 

Academic Editor

PLOS ONE